# Haemato-Immunological Response of Immunized Atlantic Salmon (*Salmo salar*) to *Moritella viscosa* Challenge and Antigens

**DOI:** 10.3390/vaccines12010070

**Published:** 2024-01-10

**Authors:** Maryam Ghasemieshkaftaki, Trung Cao, Ahmed Hossain, Ignacio Vasquez, Javier Santander

**Affiliations:** Marine Microbial Pathogenesis and Vaccinology Laboratory, Department of Ocean Sciences, Memorial University of Newfoundland, St. John’s, NL A1C 5S7, Canada; mghasemieshk@mun.ca (M.G.); ttcao@mun.ca (T.C.); ahossain@mun.ca (A.H.); ivasquezsoli@mun.ca (I.V.)

**Keywords:** Atlantic salmon, *Moritella viscosa*, hemato-immune, flow cytometry, outer membrane vesicles (OMVs), antibody

## Abstract

Winter ulcer disease is a health issue in the Atlantic salmonid aquaculture industry, mainly caused by *Moritella viscosa*. Although vaccination is one of the effective ways to prevent bacterial outbreaks in the salmon farming industry, ulcer disease related to bacterial infections is being reported on Canada’s Atlantic coast. Here, we studied the immune response of farmed immunized Atlantic salmon to bath and intraperitoneal (ip) *M. viscosa* challenges and evaluated the immunogenicity of *M. viscosa* cell components. IgM titers were determined after infection, post boost immunization, and post challenge with *M. viscosa*. IgM^+^ (B cell) in the spleen and blood cell populations were also identified and quantified by 3,3 dihexyloxacarbocyanine (DiOC6) and IgM-Texas red using confocal microscopy and flow cytometry. At 14 days post challenge, IgM was detected in the serum and spleen. There was a significant increase in circulating neutrophils 3 days after ip and bath challenges in the *M. viscosa* outer membrane vesicles (OMVs) boosted group compared to non-boosted. Lymphocytes increased in the blood at 7 and 14 days after the ip and bath challenges, respectively, in OMVs boosted group. Furthermore, a rise in IgM titers was detected in the OMVs boosted group. We determined that a commercial vaccine is effective against *M. viscosa* strain, and OMVs are the most immunogenic component of *M. viscosa* cells.

## 1. Introduction

A recognized cause of winter ulcerative disease in Atlantic salmon is *Moritella viscosa* [1]. Two main clades (‘typical’ and ‘variant’) have been recognized in this pathogen. Typical *M. viscosa* has been isolated from Atlantic salmon cultured in European countries. Variant *M. viscosa* has been isolated from Atlantic salmon reared in Canada [2,3]. There are a few published studies on skin ulcerative diseases in Atlantic salmon, but less is known about it on Canada’s east coast [4,5]. The Atlantic salmon industry attempts to prevent or control bacterial outbreaks using vaccines [6]. Although vaccination is an effective management strategy that decreases disease outbreaks and minimizes the use of antibiotics in the aquaculture industry [7], ulcerative disease is frequently reported in vaccinated Atlantic salmon in eastern Canada [8]. To develop effective vaccines, an in-depth understanding of the immune response in the target species after immunization is required. However, there are many unknown aspects regarding fish immunology, and we are still far from understanding which immune mechanisms are responsible for protecting fish against some of these pathogens [9,10,11]. Most licensed vaccines for finfish are inactivated microbes mixed with adjuvant, which contain either single or combined heat or formalin-killed pathogenic microorganisms [12,13]. An efficient vaccine stimulates the development of long-lived plasma cells that produce high-affinity antibodies and memory B-cells. Antibodies play a significant role in limiting or preventing infection and can neutralize and remove pathogens before an infection becomes severe [14]. The pathogen’s capability to grow in an iron-restricted environment led to the synthesis of Iron-Regulated Outer-Membrane Proteins (IROMP), which have been proposed as important antigens that protect against bacteria [15]. Their interaction with the host immune system makes them suitable candidates for vaccine development [16]. Additionally, bacterial outer membrane vesicles (OMVs) contain protective antigens in their structures and can be used for vaccines as well [17,18,19,20,21].

Several studies have investigated aspects of the fish immune system [22,23,24,25,26,27,28,29], and examined hematological responses in vaccinated and infected fish species [30,31]. However, there are no studies assessing cell analysis and effective immunity in Atlantic salmon following *M. viscosa* infection. Hematological analyses are commonly utilized to assess a fish’s physiological status and health [32]. For example, leukocytes are polymorphic and multifunctional immune cells and carry out various physiological and immunological tasks. They help fish adjust to various biotic and abiotic factors and protect the body from foreign substances [33]. Changes in the number of leukocytes and their differential count (e.g., lymphocytes, neutrophils, eosinophils, and monocytes) are crucial clinical indicators, as they are symptomatic of several conditions, including acute and chronic stress and pathogen exposure/infection [34].

Here, we evaluated the hematological and immune response of vaccinated and boosted farmed Atlantic salmon after *M. viscosa* challenge. Farmed Atlantic salmon immunized with ALPHA JECT micro IV (Pharmaq, Norway) containing *M. viscosa* bacterin were bath and intraperitoneally challenged with *M. viscosa*. The vaccine efficacy and immune response were examined. Also, Atlantic salmon were boosted by *M. viscosa* antigens to evaluate the antigenicity of different bacterial components (bacterin and exudates’ (complete bacterin), bacterin cell, IROMP, OMVs). IgM titers, peripherical and spleen IgM^+^ cell populations were measured in vaccinated and challenged groups. White blood cells (WBCs) showed a significant increase after the challenge, with a higher cell percentage in the OMVs boosted group compared to the non-boosted one. IgM titers increased following the vaccination and challenge in OMVs boosted group. Our results showed *M. viscosa* OMVs are highly effective at stimulating an immune response in farmed Atlantic salmon.

## 2. Materials and Methods

### 2.1. Fish Holding

Vaccinated farmed Atlantic salmon (~350 ± 50 g) provided by Cook aquaculture industry were held at the Ocean Sciences Center (Memorial University of Newfoundland). Fish were kept and fed according to the standard protocols [8], and ethical procedures #18-01-JS, #18-03-JS, and biohazard license L-01 were implemented for the study.

### 2.2. Infection Trials

#### 2.2.1. Inoculum Preparation

One colony of *M. viscosa* J311 (ATCC BAA-105) was cultivated in 3 mL of Trypticase Soy Broth (TSB, Difco) supplemented with 2% NaCl and set in a drum roller (TC7, New Brunswick, NJ, USA) at 15 °C for 24 h. Then, 300 µL of the culture was mixed with 30 mL of TSB 2% NaCl. Following, it was kept at 15 °C incubator for 24 h with aeration at 180 rpm. Bacterial proliferation was measured by Genesys 10 UV spectrophotometer (Thermo Spectronic, Thermo Fisher Scientific, Waltham, MA, USA) up to an OD ~0.7. Then, the bacterial culture was subjected to centrifugation at 4200× *g* for 10 min to isolate the sediment. The pellet was washed twice with filtered-sterilized seawater (0.22 μm) and resuspended in 300 µL of filtered-sterilized seawater, and its concentration was determined by plating method [35,36,37].

#### 2.2.2. Challenge Assay in Vaccinated Atlantic Salmon

Atlantic salmon were vaccinated with ALPHA JECT micro IV when they reached a weight of approximately 60–80 g on 16 October 2020, six months before the challenge. This procedure was performed in the AQ3 biocontainment unit at the Cold-Ocean and Deep-Sea Research Facility (CDRF) at MUN. Fish were transferred to the AQ3-CDRF unit and acclimated for one week at 10 °C. All of the 240 immunized Atlantic salmon (~350 ± 50 g) were evenly distributed into six 500 L tanks containing 40 fish each. Fish were divided into three groups (Appendix A). Two tanks containing 80 fish were anesthetized with 50 mg/L MS-222 (Syndel Laboratories, Vancouver, BC, Canada) and intraperitoneally (ip) injected with 100 µL (1 × 10^6^ CFU/dose) of *M. viscosa*, and two other tanks were bath challenged (1 × 10^6^ CFU/mL) for 30 min. The final two tanks were not infected and were used as negative control groups. The susceptibility of Atlantic salmon was evaluated, and mortality rates were recorded daily up to 30 days post challenge (dpc).

### 2.3. Flow Cytometry

We performed flow cytometry, which is the most common technique used for single-cell analysis and isolation [38], to quantify the different blood cell populations in the blood and IgM^+^ cells in the spleen. Briefly, three fish from each group were netted at 3, 7, and 14 dpc and euthanized instantly with an overdose of MS222 (400 mg/L). Spleen and blood samples were aseptically collected. Spleens were homogenized using a 100 μm mesh strainer and resuspended in FACS media. Blood samples were heparinized (100 mg/mL. Pfizer Inc, NY, USA). All suspensions were preserved on ice until performing flow cytometry. The procedures and analyses were conducted according to previously described methods with modifications [39]. Briefly, fresh blood or spleen cell suspensions (20 μL) were diluted in FACS media (1:100 μL). Then, they were stained with 40 μL of 3,3 dihexyloxacarbocyanine (DiOC6) (1 μg/mL), 2 μL of Texas red-IgY anti-salmon IgM fresh solution. After staining, blood cell populations and IgM^+^ in the spleen were analyzed by a BD FACS Aria II flow cytometer (BD Biosciences, San Jose, CA, USA) and BD FACS Diva v7.0 software.

### 2.4. Immune Confocal Microscopy

Three healthy fish were euthanized with an overdose of MS222 (400 mg/L), and their spleens were aseptically collected, homogenized using a 100 µm mesh strainer, and resuspended in FACS media (PBS; 0.1% fetal bovine serum (FBS; Gibco, NY, USA)). Biotinylated chicken IgY anti-salmon IgM (1 mg/mL, Somru BioSciences, Charlottetown, PE, Canada) was labeled with Texas red-avidin (2.5 mg/mL, Thermo Fisher Scientific, MA, USA) by mixing in a 1:1 volume for 30 min at room temperature. Then 50 µL of the spleen cell suspensions were gently mixed with 3 µL of Texas red-IgY anti-salmon IgM, and 1 µL of 4,6-diamidino-2-phenylindole (DAPI, 5 mg/mL concentration) (Thermo Fisher Scientific, MA, USA) in a 1.5 mL centrifuge tube. A 3 µL aliquot of the stained cell suspension was added onto mountain media (Thermo Fisher Scientific, MA, USA) on a microscope slide, covered with a cover slide, and sealed. These samples were visualized by confocal microscope (Nikon AX/AX R, Long Island, NY, USA) for image acquisition. This study was conducted to verify the flow cytometry results.

### 2.5. IgM Titer Determination Using Indirect Enzyme-Linked Immunosorbent Assay (ELISA)

Six fish from each group were netted at 3, 7 and 14 dpc and anesthetized with 50 mg/L MS222. Blood samples were taken and centrifuged at 4200× *g* for 5 min at room temperature and preserved at −80 °C between 8 and 12 h. The samples were thawed in ice, and the complement system was inactivated by heating the samples to 56 °C for 30 min. Then, lipids were removed by adding 100 μL of chloroform (Sigma-Aldrich, St. Louis, MO, USA) and centrifuged at 4200× *g* for 10 min at room temperature. Next, the supernatant of each sample was collected and stored at −80 °C until analysis. Indirect ELISAs were conducted to quantify serum antibodies against *M. viscosa* according to published methods [40,41], with modifications. Briefly, the ideal antigen concentration (Formalin-killed *M. viscosa* cell antigen) to react with salmon serum was determined by a checkerboard titration method [42]. The anti-*M. viscosa* antibodies in the serum were measured at six distinct antigen concentrations (4, 2, 1, 0.5, 0.25, and 0.125 μg/mL) to interact with the pooled serum of Atlantic salmon (1:2 to 1:64). Antigen concentrations were coated into polystyrene 96-well microtiter plates (Thermo Scientific, MA, USA), and they were then incubated at 4 °C overnight. Following, the wells were blocked with 150 μL of ChonBlock™ (Chondrex Inc, Woodinville, WA, USA) for 1 h at 37 °C, washed three times with PBS-Tween, and then incubated with various dilutions of control and *M. viscosa* infected salmon. Based on the results of these assays, 4 μg/mL was chosen.

Finally, 96-well microtiter plates were coated with 100 μL (4 μg/mL) of *M. viscosa* in coating buffer (0.015 mM Na_2_CO_3_; 0.035 mM NaHCO_3_; pH 9.8) and were kept overnight at 4 °C. The antigen-coated plates were washed with 100 μL of PBS 0.1% Tween-20 (PBS-T) three times and blocked with 150 µL of Chondrex buffer for 1 h at 37 °C. Then, the 96-well plates were washed with 100 μL of PBS-T 0.1% three times, and 100 µL of Atlantic salmon serum from the infected or control group were added to the first well row, serially diluted 2-folds until the last row, and incubated at 37 °C for 1 h. Next, the wells were washed with 100 μL of PBS-T five times, and 100 µL of IgY anti-salmon IgM diluted with PBS-T (1:10,000) was added and the plates were incubated for 1 h at 37 °C. Afterward, the plates were washed five times with 100 μL of PBTS, and incubated with 100 µL of Streptavidin-HRP (Southern Biotech, Birmingham, AL, USA) diluted with PBST (1: 10,000) for 1 h at 37 °C. Following, the plates were washed three times with PBST. Finally, 50 µL of TBM buffer (3,3′, 5,5-tetramethylbenzidine; (Thermo Scientific, MA, USA) was added to each well and were incubated for 15 min. The chemical interaction was stopped by adding 50 µL of 2M H_2_SO_4_, and absorbance was determined at 450 nm using a microplate reader (SpectraMax M5 Multi-Mode Microplate Reader, Molecular Devices, Sunnyvale, CA, USA). To find the ideal cut-off, the specific antibody titer was determined by taking the maximum serum dilution, at which the O.D values began to rise as the dilution factor rose. This number was then normalized to a logarithmic base two scale [42].

### 2.6. M. visosa Antigenicity in Farmed Atlantic Salmon

#### 2.6.1. Vaccine Preparations 

Four different vaccines (bacterin cell, bacterin with exudates’ (complete bacterin), IROMP, OMVs) were used in this study. For preparing the bacterin cell, one colony of *M. viscosa* was cultivated in a tube containing 3 mL of TSB supplemented with 2% NaCl for 48 h at 15 °C with aeration (180 rpm). Then, 500 µL of the tube’s content was transferred to a 250 mL flask containing 50 mL of TSB 2% NaCl and grown at 15 °C up to an OD ~0.6–0.7 at 600 nm, and these bacteria were harvested by centrifugation (4200× *g*, 10 min, 4 °C) and washed two times with seawater and resuspended in 500 µL of seawater. Bacterial cell count was measured after 72 h incubation at 15 °C [37]. After counting, the bacterial cells were inactivated by adding 6% (*v*/*v*) of formalin (Sigma Chemical, St. Louis, MO, USA) and incubated at 4 °C for 48 h. Formaldehyde was removed by centrifugation, the cells were washed three times with PBS 1X, and then the cells were dialyzed (Spectrum™ Spectra/Por™ dialysis membrane; 12–14,000 Dalton molecular weight cut-off; Thermo Fisher, USA) in 1 L PBS 1X at 4 °C for 24 h with agitation in an orbital shaker [43]. *M. viscosa* inactivation was confirmed by plating onto TSA 2% NaCl for 24 h at 15 °C. The final concentration of bacterin (4.1 × 10^8^ CFU/mL) was determined by a flow cytometer (Appendix A) [44,45]. *M. viscosa* bacterin was kept at 4 °C until use. For vaccine preparation, 10% carbigen carbomer-based (Carbopol 934P) adjuvant were mixed with bacterin (4.1 × 10^8^ CFU/mL) and buffered to pH 7 according to the manufacturer’s protocols (MVP Adjuvants^®^, Phibro Animal Health Corporation, Teaneck, NJ, USA).

For preparing bacterin with exudates (complete bacterin), *M. viscosa* was grown in a flask containing 50 mL TSB supplemented with 2% NaCl up to an OD ~0.6–0.7 at 600 nm, as previously described. Then, 5 mL of *M. viscosa* was grown in 500 mL of TSB 2% NaCl up to an OD ~0.9 at 600 nm. The bacteria were inactivated with 0.4% formaldehyde for 3 days with gentle shaking at room temperature. Then, the final concentration was calculated by flow cytometer as outlined previously (Appendix A), and formalin-killed bacterin (4.2 × 10^7^ CFU/mL) was mixed with 10% carbigen adjuvant and kept at 4 °C for immunization.

Laboratory preparation of *M. viscosa* outer-membrane proteins (OMPs), IROMP and OMVs, was carried out by established protocols [46,47,48,49]. Then, 100 µL of IROMP containing 50 µL of IRON (1504.4 µg/mL) and 50 µL *M. viscosa* OMP (1440 µg/mL) were added to 4450 µL of PBS and mixed with 450 µL of carbigen adjuvant, according to the manufacturer’s instructions, up to a concertation of 1 mg/mL (1:1 OMPs and IROMPs). For *M. viscosa* OMVs vaccines, 50 μL of OMVs (1546.3 µg/mL) were added to 4.5 mL of PBS and mixed with 450 μL of carbigen adjuvant.

#### 2.6.2. Antigenicity of *M. viscosa* Components in Farmed Atlantic Salmon

This experiment was performed at the Dr. Joe Brown Aquatic Research Building (JBARB) in the Ocean Science Center, MUN. A total of 270 vaccinated Atlantic salmon with an average weight of ~380 ± 50 g were Passive Integrated Transponder (PIT)-tagged and divided equally into six 500 L tanks, each containing 45 fish. Atlantic salmon were anesthetized with 50 mg/L of MS222 and individually injected with 100 μL of the vaccine preparation (e.g., *M. viscosa* bacterin, *M. vicosa* cell, IROMP, OMVs). In each tank, 45 fish were intraperitoneally boosted with their respective vaccine, and one group were injected with PBS mixed with carbigen adjuvant and used as the control group (Appendix A). After immunization, blood samples from 6 fish were taken from each treatment every two weeks (2, 4, 6, and 8). The serum of each sample was collected and stored at −80 °C until IgM titer was determined by indirect ELISA.

#### 2.6.3. *M. viscosa* Challenge in Atlantic Salmon

Atlantic salmon boosted with 4 vaccines and PBS mixed with carbigen adjuvant were transferred from JBARB to 6 tanks in the AQ3-CDRF and acclimated for 1 week at 10 °C under the described optimal conditions. Twelve weeks after being boosted, fish were challenged with *M. viscosa*. Briefly, two tanks containing 90 fish with 6 treatments were bath challenged with *M. viscosa* (1 × 10^6^ CFU/mL) for 30 min without seawater flow through. The seawater flow through was reestablished after the challenge. The fish in two tanks containing different treatments were anesthetized with 50 mg/L MS222 (Syndel Laboratories, Vancouver, BC, Canada) and individually injected with 100 µL (1 × 10^6^ CFU/dose) of *M. viscosa*. Two tanks were not challenged and remained as the control groups (Appendix A). Blood and spleen samples were aseptically collected at 3, 7, and 14 dpc to analyze the cell populations. IgM titers in the blood were detected using indirect ELISA. The mortality rates were recorded 30 days post challenge (Appendix A). 

### 2.7. Statistical Analysis

The research data were analyzed using GraphPad Prism 10 (California, CA, USA). An arcsin (survival rate ratio) function was utilized to calculate fish survival rates. Shapiro–Wilk test was conducted to assess the normality of the data. All data are not normally distributed, and one-way ANOVA (non-parametric) followed by a Kruskal–Wallis’s test was utilized to determine differences (*p* ≤ 0.05) between cell populations. Two-way ANOVA (Dunnett’s multiple comparison tests) was performed to analyze the ELISA data. Asterisk (*) indicated considerable differences (*p* ≤ 0.0001) among the reciprocal titers.

## 3. Results

### 3.1. Susceptibility of Vaccinated Farmed Atlantic Salmon to M. viscosa Challenge

Mortality was not observed in the control (non-infected) group, in contrast, at 30 dpc, mortality in the bath and ip-challenged animals was 2.82% and 5.64%, respectively (Figure 1).

### 3.2. Cell Populations in the Challenged Atlantic Salmon

Blood cell populations were analyzed by flow cytometry (Appendix A). The IgM^+^ (B cell) population was defined in the spleen (Appendix A).

The cell populations were divided into two parts. Red blood cells (RBCs) were shown in P1. WBCs were highlighted in P2 (Appendix A). Then, P2 was divided into three distinct populations (Appendix A). P3 exhibited IgM^+^ (B cells), P4 showed neutrophils and basophils, and P5 indicated monocytes (Appendix A). Neutrophils and basophils increased significantly at 3 days post ip and bath challenge as compared to the non-infected salmon. Also, lymphocytes were considerably higher than the control group at 7 and 14 days post ip and bath challenges, respectively. Monocytes did not show meaningful differences between the groups (Figure 2).

We found no significant differences between the spleen IgM^+^ cell populations of control and challenged groups at 3 and 7 dpc. However, at 14 dpc, IgM^+^ cell population reached 15.4% in the ip challenged group and was significantly higher compared to the non-infected animals (Figure 3).

### 3.3. Confocal Microscopy Analysis

Immunostaining of white blood cells (WBCs) using DAPI, DiOC6, and Texas red IgY anti-salmon IgM revealed that peripheral IgM^+^ leukocytes were present in the spleen (Figure 4). This result validated the flow cytometry findings.

### 3.4. IgM Titer Levels in the Challenged Atlantic Salmon

IgM titers did not change with time in the control group (non-infected) salmon. However, they were higher at 14 dpc in the ip-challenged group. Bath-challenged animals showed slightly higher IgM titers than the control group at all the examined time points. However, this difference was not significant (Figure 5).

### 3.5. Detection of Antibody after Booster Dose by Prepared Vaccines

Immunized fish boosted with the OMVs vaccine had the highest IgM levels at all the examined time points, and this was significantly greater than in the control fish at all time points. In contrast, the IROMP-boosted group only had greater IgM titers at 4, 6, and 8 weeks post boost immunization compared to the non-boosted group, while that of the bacterin cell-boosted group was only higher than the control group at 6 weeks following the boost. Finally, the salmon boosted with the complete bacterin did not have IgM titers higher than those measured in the non-boosted (i.e., PBS adjuvant injected group) (Figure 6).

### 3.6. Cell Population after Challenge in the Boosted Groups

Our findings showed that neutrophils and basophils increased in the OMVs boosted group at 3 dpc (ip and bath) compared to the non-boosted one (Figure 7 and Figure 8). Furthermore, lymphocytes were statistically higher in the OMVs boosted animals compared to the non-boosted group at 7 and 14 days post ip and bath challenge, respectively. A slight rise of leukocytes was observed in other boosted groups compared to the non-boosted; however, the non-parametric test revealed no significant differences between them (Figure 7 and Figure 8). IgM^+^ (B cell) increased in the spleen and peaked at 14 dpc in the OMVs-boosted group but was not statistically different from non-boosted animals (Figure 9).

### 3.7. Detection of Antibody Levels after Challenge in the Boosted Groups

The antibody titer slightly increased from 3 to 14 days post challenge (dpc) (Figure 10A,B). The OMVs vaccine reached the highest antibody level at all the studied time points and was statistically higher at 14 dpc than the non-boosted animals (Figure 10B). The IROMP-boosted group displayed slightly higher antibody titer than the bacterin cell, complete bacterin, and control groups. However, statistical analysis indicated no significant difference between them. Antibody levels were slightly higher in the ip-challenged animals compared to the bath-challenged group. Also, no mortality was observed in the boosted animals until 30 dpc. This investigation indicated a positive correlation between survival proportions and antibody titers. All the vaccines induced the fish immune response, and the survival rate was 100% in all the treatments after the challenge (Appendix A).

## 4. Discussion

Winter ulcer disease is mainly but not solely caused by *M. viscosa* [1,3,50]. Winter ulcers cause mortality, downgrading at slaughter, and animal welfare concerns [51]. Bacterin-based vaccines are usually used to control bacterial disease in the Atlantic salmon industry [52]. This study assessed farmed Atlantic salmon’s immunological response and susceptibility to *M. viscosa* challenge. ALPHA JECT-vaccinated Atlantic salmon were bath- and ip-challenged. Only 2.82% and 5.64% mortality were recorded in the bath and ip challenge, respectively. All the control (non-infected) groups survived, and the survival portions in the bath- and ip-challenged animals were 97.18% and 94.36%, respectively. The high survival percentage showed an appropriate immune response of infected fish and confirmed the effectiveness of the ALPHA JECT micro IV vaccine in protecting Atlantic salmon from *M. viscosa* challenge (Figure 1). The virulence of this *M. viscosa* strain has been previously verified in Atlantic salmon [8]. This result was expected since Norwegian strains are used as a vaccine component, such as the *M. viscosa* type strain used in this study. Efficacy against local strains of Atlantic Canada might be valuable for further study.

Other studies showed the efficacy of vaccines against *M. viscosa*. Vaccination against *M. viscosa* significantly reduced the clinical effects of this pathogen, including mortality and skin ulceration. Relative protection reached 91% compared to saline controls and 65% compared to the vaccine formulation lacking *M. viscosa* antigen. Exposure to *M. viscosa* antigens generally resulted in protective immunity. The survivors’ group did not exhibit the clinical symptoms of winter ulcer disease [3]. This result was consistent with our findings. In this study, no clinical signs of winter ulcer disease were observed after the challenge in the vaccinated Atlantic salmon. 

Rozas-Serri et al. (2022) reviewed the circulating blood cells of farmed Atlantic salmon. They indicated lymphocytes had the largest proportion of white blood cells (WBCs) in all the age ranges in farmed Atlantic salmon [53], consistent with our current study results. Lymphocytes comprised most of the WBC populations, followed by neutrophils and monocytes (Figure 2, Figure 7 and Figure 8).

Hematological analysis showed an increase in the leukocyte population after *M. viscosa challenge*. Neutrophils and basophils increased significantly at 3 days post ip and bath challenges (Figure 2). This suggests a leukocyte adhesion cascade [54,55], crucial for early defense against infection [56].

Lymphocytes considerably increased at 7 and 14 days ip and bath challenge. Lymphocytes, including B cells and T cells, have essential roles in the adaptive immune response [57]. Our results showed that lymphocytes peaked earlier in the ip challenge compared to the bath route. This relatively early peak might indicate that adaptive immune response was triggered more rapidly in the ip challenge. The immune response in the ip route may have been accelerated, leading to the earlier recruitment and activation of lymphocytes. The delayed peak (at 14 days) in the bath challenge suggested that adaptive immune response took longer to develop in the bath route. An increase in leukocytes after bacterial infections was observed in other fish species [30,58,59,60], which is consistent with our finding after infection with *M. viscosa* (Figure 2).

IgM has been associated with mucosal and systemic immunity [29,61,62]. Teleost fish membrane-bound IgM^+^ (B lymphocytes) are a B cell subset implicated in innate and adaptive immune responses [63]. In this study, IgM^+^ (B cell) increased after the challenge and was statistically different from the non-infected animals at 14 days ip challenge in the spleen. B cells secrete antibodies against invading infections and are at the center of humoral immunity [64]. B lymphocytes that have undergone terminal differentiation and can secrete antibodies are plasma cells [64]. Lymphocytes were not statistically higher at 14 days ip challenge in the blood; however, a great volume of antibodies was observed in the serum (Figure 5). This finding suggested that B cells may differentiate into plasma cells and produce antibodies.

Rising the IgM after infection was reported in the spleen [65]. In our study, IgM increased at 2 weeks following the ip challenge (Figure 3). The early appearance of IgM^+^ in the spleen might suggest that vaccination quickly induced the immune response after infection. The pre and post infection serum IgM titers had been reported to be significantly greater in vaccinated fish compared to the unvaccinated. This result indicated an active immune response and phagocytic activity [31], which matched our results in this study (Figure 6 and Figure 10B).

Most licensed aquaculture vaccines are inactivated pathogenic microbes mixed with adjuvant, which contain either single or combined microorganisms [12,13]. These vaccine preparations contain not only immune protective antigens but also immune suppressive molecules and identify the most immunogenic bacterial components that can contribute to improving vaccine efficacy. This study used four different vaccines to boost the animals and reviewed the immunogenic response of *M. viscosa* antigen in vaccinated Atlantic salmon after the challenge of determining the immunogenicity of these vaccine preparations.

In this investigation, one group of salmon was boosted with the bacterin vaccine containing killed *M. viscosa* and exudates, such as the inactivated commercial vaccines that are commonly used in aquaculture. Also, one group of salmon was boosted with a bacterin cell vaccine containing only *M. viscosa* cells.

IROMP, a vaccine utilized in this investigation, is the combination of iron and the OMPs of *M. viscosa*. Gram-negative bacteria’s OMPs represent a significant portion of the cell surface. They are crucial structural elements of the membrane, acting in virulence-related processes like adhesion, invasion of host tissues, and reducing the host immune response. They are also highly antigenic [16,66]. Since they interact with the host immune system, they are important vaccine candidates [16].

OMVs, a vaccine preparation used in this study, are spherical lipid vesicles released from the outer membranes of Gram-negative bacteria to facilitate communication among the organisms as they grow in various environments and control the host immunological response [67,68]. OMVs are composed of OMPs, phospholipids, peptidoglycan, lipopolysaccharides (LPS), proteins, nucleic acids, ion metabolites, and signaling molecules [67,69]. OMVs have been classified and used in many ways. They contain adhesion, autolysins, cytotoxins, virulence factors, nutrition acquisition, stress responses, and toxin delivery that elude host defense mechanisms, among many more biomolecules [67]. In primary infection, OMVs are critical for delivering highly virulent factors to host cells, where they dramatically degrade the host cells’ enzymes and cause cell death. A crucial candidate for a new vaccine formulation is bacterial outer membrane vesicles carrying antigens. They efficiently phagocytize antigen-presenting cells due to the surface-associated antigens they transport. They also bring a variety of PAMPs (pathogen-associated molecular patterns) that trigger and support immune system responses [70]. It supported our findings in this study, which demonstrated a high capacity of OMVs to induce an immunological response in Atlantic salmon (Figure 6, Figure 7 and Figure 8).

OMVs have demonstrated the specific nature of the protective antibody and significant antigens that induce an adaptive memory immune response and possess self-adjuvant characteristics [71]. Our findings were in line with this. According to our research, adaptive immune response quickly triggered following the challenge, and OMVs-boosted group had higher lymphocytes and IgM titers than the other treatment (Figure 7, Figure 8 and Figure 10B).

It has been demonstrated that OMVs can cause immunological response and protect fish [72]. *P. salmonis* OMVs cause the generation of IgM in Atlantic salmon. IgM was produced against *P. salmonis* proteins in the serum taken from immunized fish after 14 days [73]. This was consistent with our findings in Atlantic salmon boosted with *M. viscosa* OMVs. IgM was detected at 14 days post boost immunization (Figure 6).

The OMVs vaccines have been evaluated in developing fish vaccines against *Francisella noatunensis* in zebrafish, *P. salmonis* in salmonids, and *V. anguillarum* in Japanese flounder [49,73,74]. These findings suggested that OMVs effectively induce stable immune responses, confer protection against bacteria, and are efficient antigens to produce vaccines [75,76]. In this study, OMVs were extracted from *M. viscosa* and used as the vaccine component. This is the first research that utilized the OMVs vaccine to boost the farmed Atlantic salmon against *M. viscosa*. The results of our experiment showed *M. viscosa* OMVs are the most immune-stimulating antigens compared to other vaccine elements in Atlantic salmon (Figure 6, Figure 7, Figure 8 and Figure 10B).

Our findings indicated that antibody titer in the boosted group with OMVs significantly differed from the control group at all the examined time points (2, 4, 6, and 8 weeks). Antibody titers peaked at 6 weeks in all the treatments and reached the highest in the OMVs boosted group. The IROMP and bacterin cell antibody levels were in second and third place, respectively, and they differed significantly from the control group at 6 weeks post boost immunization (Figure 6). Several studies were conducted on Atlantic salmon and they reviewed antibody titer by ELISA after vaccination. Romstad et al. (2012) indicated that Atlantic salmon immunized with A-layer positive vaccines demonstrated an increased mean antibody response with rising antigen dose [77]. According to Liu et al. (2020), all the immunized groups had noticeably higher antibody levels post vaccination in comparison to the control group [78]. Rising the antibody was consistent with what we observed in this study after boosting (Figure 6).

Also, in other fish species, significant rise of antibody was observed after vaccination. Japanese flounder was vaccinated against *P. fluorescens* and *A. hydrophila*. In all cases, specific antibodies were found at 5 weeks post vaccination and persisted for up to 8 weeks post vaccination [79]. Similar results were obtained in this study in Atlantic salmon. At 6 weeks post boost immunization, a significant difference in IgM titers was observed between the boosted groups (OMVs, IROMP, bacterin cell) and non-boosted animals. The IgM titers remained relatively high in OMVs and IROMP boosted animals compared to the non-boosted group to the end of the experiment at 8 weeks post boost immunization (Figure 6). A considerable rise in IgM titers after the challenge was reported in rainbow trout [80], similar to our results. High IgM titers were detected in the boosted animals compared to the non-boosted group (Figure 6). IgM titers were noticeably higher in the OMVs boosted animal at 14 days following ip challenge (Figure 10B). Villumsen et al. (2012) reported that the experimental vaccine produced noticeably greater IgM titers in each of the three intervals between immunization and challenge compared to the unvaccinated controls [81], which is similar to our results (Figure 6 and Figure 10B).

Hematological analysis revealed a considerable rise in the neutrophils and basophils in the OMVs boosted group compared to the non-boosted fish at 3 dpc (Figure 7 and Figure 8). Lymphocytes increased at 7 and 14 days post ip and bath challenge, respectively, in the OMVs boosted group (Figure 7 and Figure 8). A tendency of the rise was observed in the IgM^+^ cells in the spleen of OMVs-boosted salmon compared to the non-boosted group (Figure 9). *M. viscosa* OMVs might contain several immunogenic molecules and perhaps fewer immunosuppressors, thus, boosted salmon may have a faster activation of immune cells, including neutrophils, upon *M. viscosa* infection. Further studies on the *M. viscosa* OMVs protein profile might provide an antigenic profile for vaccine development against *M. viscosa*.

## 5. Conclusions

This is the first comprehensive study using flow cytometry to analyze the hematological cells of vaccinated and boosted farmed Atlantic salmon before and after *M. viscosa* challenge. This investigation evaluated the immune response of Atlantic salmon and reviewed the antibody titers after vaccination and challenge. Our research showed that the booster dose of OMVs vaccine had a strong capacity to trigger the farmed Atlantic salmon’s immune response following immunization and challenge with *M. viscosa*. Leukocytes were statistically higher in the OMVs boosted group, which indicated the appropriate haemato-immune response of the vaccine to control the infection. *M. viscosa* OMVs could be used as vaccines, heightened antibody response and potentially enhanced immune cell activity. These advantages contribute to the overall effectiveness of the vaccination strategy in conferring immunity against the specific pathogen.

## Figures and Tables

**Figure 1 vaccines-12-00070-f001:**
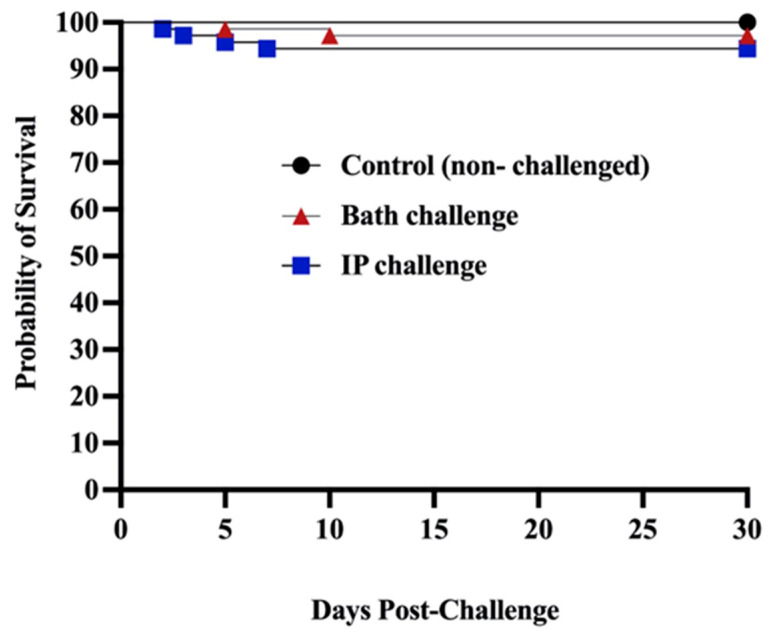
Survival percentages of ALPHA JECT vaccinated farmed Atlantic salmon after being challenged with *M. viscosa*. The experimental fish were ip and bath challenged with 10^6^ CFU/dose and 10^6^ CFU/mL of *M. viscosa*, respectively. The control group was not infected.

**Figure 2 vaccines-12-00070-f002:**
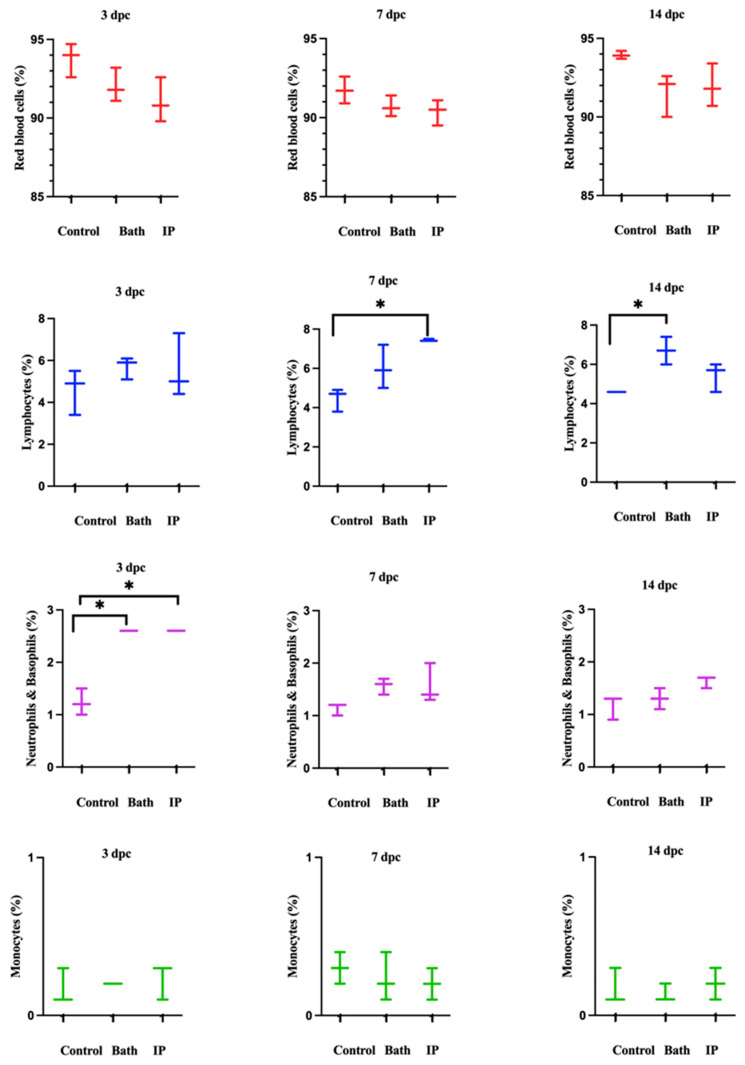
Atlantic salmon blood cell populations identified by flow cytometry at 3, 7, and 14 dpc with *M. viscosa*. Differences were identified using one-way ANOVA (non-parametric) followed by Kruskal–Wallis post hoc tests at each sampling point. Asterisks (*) demonstrate a significant difference between groups (*p* ≤ 0.05).

**Figure 3 vaccines-12-00070-f003:**
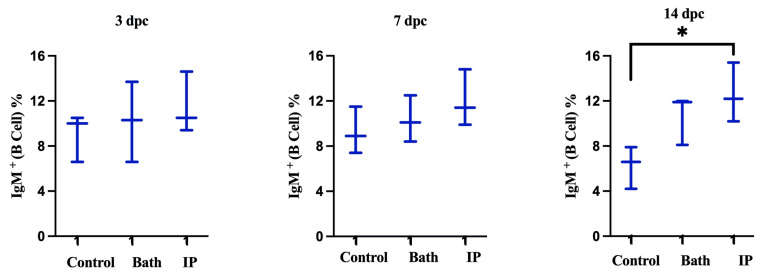
Spleen IgM^+^ (B cells) detected by flow cytometry in ALPHA JECT-vaccinated Atlantic salmon at 3, 7, and 14 dpc with *M. viscosa*. Differences were identified by performing one-way ANOVA (non-parametric) followed by Kruskal–Wallis post hoc tests. At each time point, challenged groups are compared versus control (non-infected) fish. Asterisk (*) demonstrates a significant difference (*p* ≤ 0.05) between groups.

**Figure 4 vaccines-12-00070-f004:**
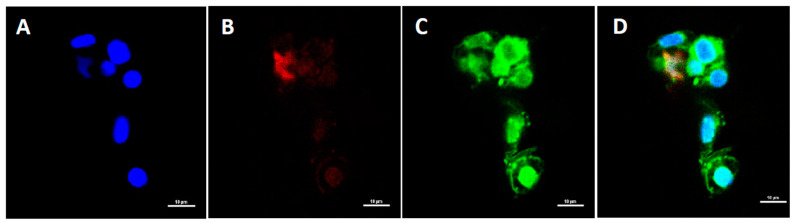
Confocal microscopy of Atlantic salmon leukocytes from the spleen. (**A**) Cell nuclei were stained blue with DAPI; (**B**) IgM^+^ cells with red fluorescence were visible following staining with chicken anti-salmon-IgM/IgY and avidin-Texas Red (TX-R); (**C**) green fluorescence indicates cells that were stained with FITC; (**D**) overlay of all the used colors in confocal imaging.

**Figure 5 vaccines-12-00070-f005:**
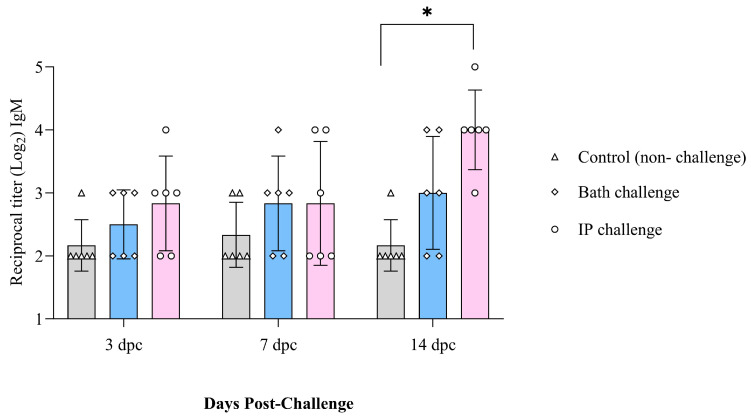
IgM serum titers in Atlantic salmon at 3, 7, and 14 dpc with *M. viscosa* determined by indirect ELISA. Significant differences were identified using two-way ANOVA followed by Dunnett’s multiple comparisons test. The asterisk (*) indicates a significant difference (*p* ≤ 0.0001) between the treatment and the control group at a particular sampling point.

**Figure 6 vaccines-12-00070-f006:**
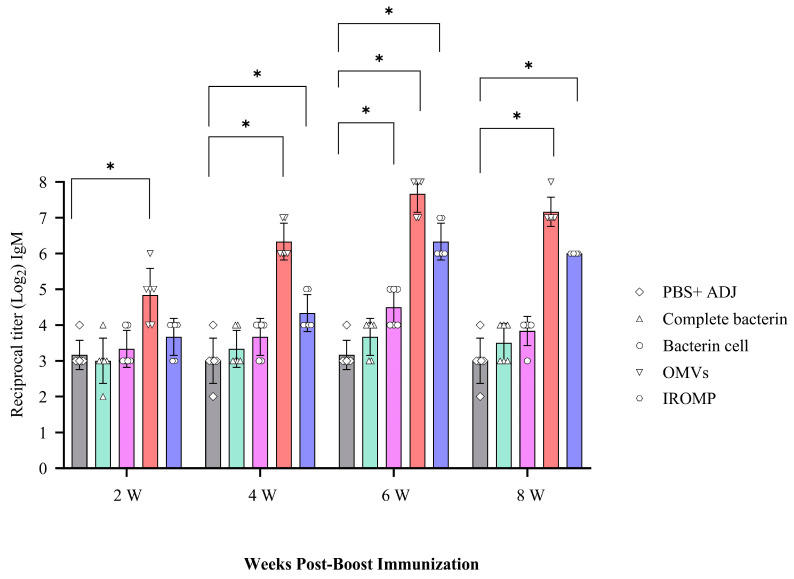
IgM serum titers in Atlantic salmon at 2, 4, 6 and 8 weeks post boost immunization determined by indirect ELISA. Significant differences were identified using two-way ANOVA followed by Dunnett’s multiple comparisons post hoc test. Asterisks (*) demonstrate meaningful differences (*p* ≤ 0.0001) between the reciprocal titers.

**Figure 7 vaccines-12-00070-f007:**
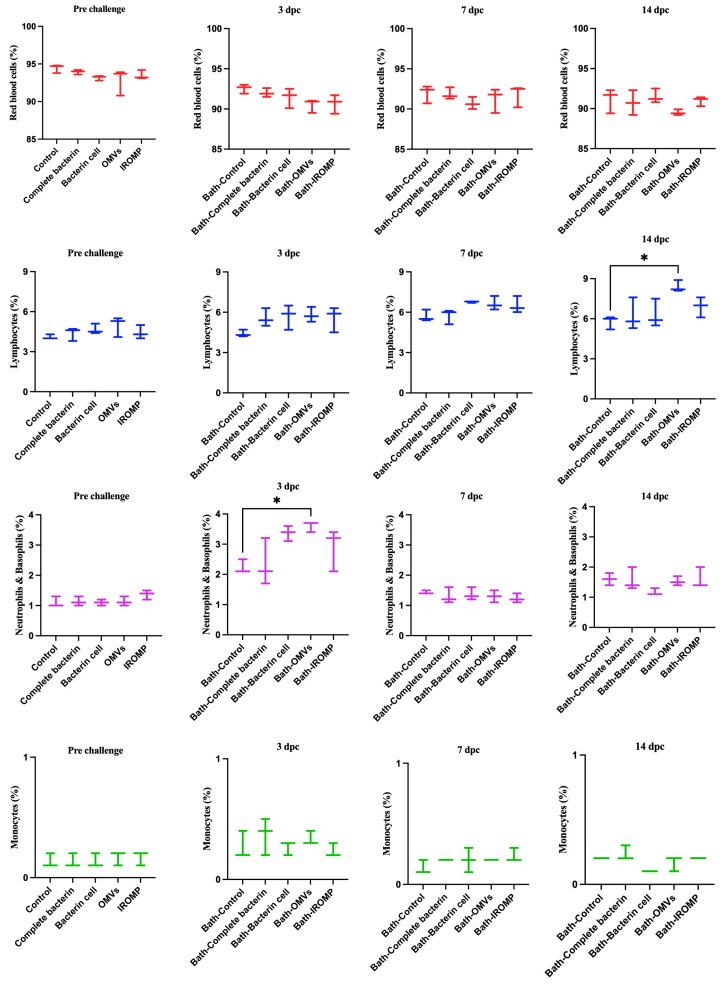
Blood cell populations in boosted Atlantic salmon analyzed by flow cytometry at 3, 7 and 14 days post bath challenge with *M. viscosa*. Significant differences were identified by performing one-way ANOVA (non-parametric) followed by Kruskal–Wallis post hoc tests. Asterisks (*) reveal a meaningful difference (*p* ≤ 0.05) between experimental (i.e., boosted) and control fish (i.e., given a PBS injection with adjuvant).

**Figure 8 vaccines-12-00070-f008:**
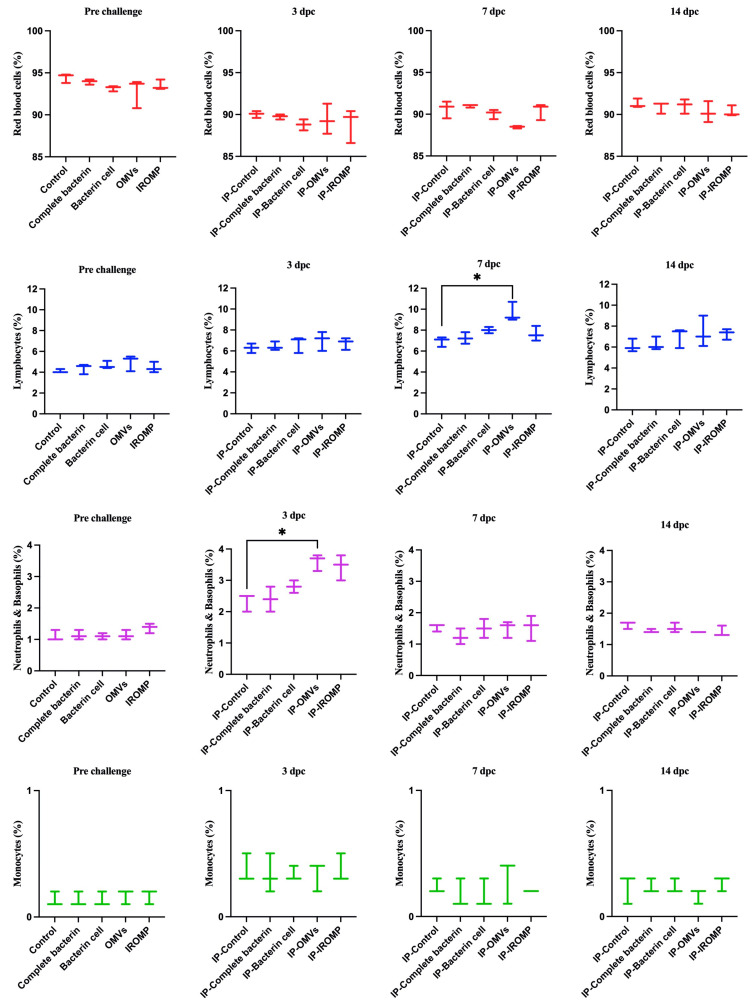
Blood cell populations in boosted Atlantic salmon identified by flow cytometry at 3, 7, and 14 days post ip challenge with *M. viscosa*. Significant differences were identified using one-way ANOVA (non-parametric) followed by Kruskal–Wallis post hoc tests. Asterisks (*) indicate a significant difference (*p* ≤ 0.05) between experimental (i.e., boosted) and control fish (i.e., given a PBS injection with adjuvant).

**Figure 9 vaccines-12-00070-f009:**
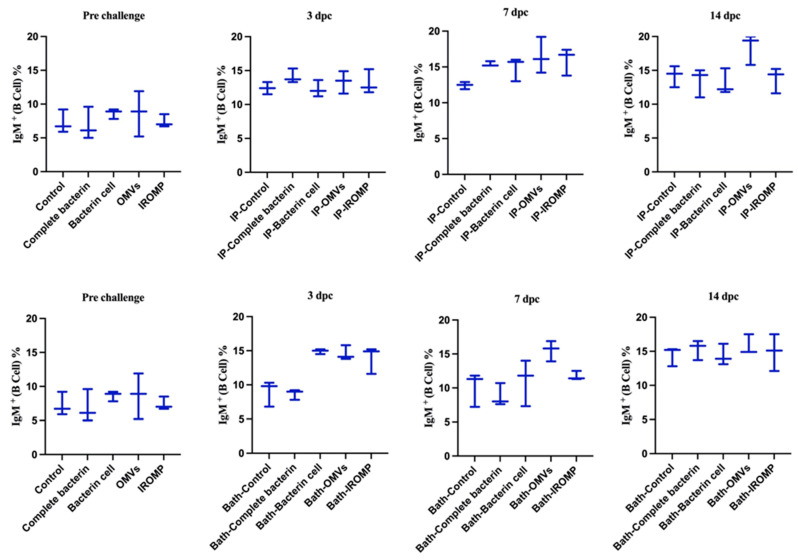
IgM^+^ (B cells) in the spleen of boosted Atlantic salmon 3, 7 and 14 days after ip and bath challenges with *M. viscosa.* No significant differences (*p* ≤ 0.05) were identified between the non-boosted and boosted treatments after performing one-way ANOVA (non-parametric) followed by Kruskal–Wallis post hoc tests.

**Figure 10 vaccines-12-00070-f010:**
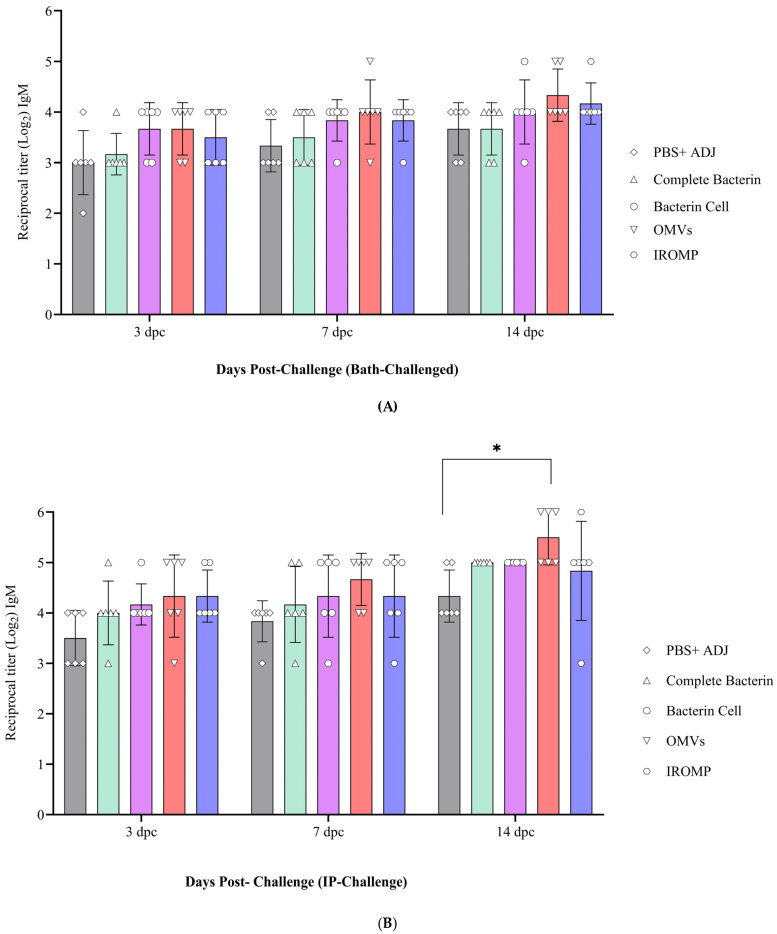
Serum IgM titers in Atlantic salmon at 3, 7 and 14 days (**A**) post bath and (**B**) post ip challenges. Two-way ANOVA (Dunnett’s multiple comparisons tests) were used to detect significant differences. The asterisk (*) demonstrates a significant difference (*p* ≤ 0.05) between the boosted and non-boosted fish.

## Data Availability

Data is contained within the article and Appendix A.

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
