# Peer review of "Haemato-Immunological Response of Immunized Atlantic Salmon (Salmo salar) to Moritella viscosa Challenge and Antigens"

_vaccines, 2024, doi:10.3390/vaccines12010070_

Round 1

Reviewer 1 Report

Comments and Suggestions for Authors

The utilization of flow cytometry to analyse vaccinated Atlantic salmon response to infection makes this work a novelty on the field. It showcases a fresh perspective on the analysis of effectiveness of vaccination in different portions of the cells. This has the potential to open new avenues for future research in the domain. Overall, your manuscript makes a good contribution to the existing knowledge, and I believe that the incorporation of these techniques adds significant value to the field. I appreciate the thoroughness of your work and the clarity with which you present your findings.

I would like to see the results organized differently, and/or with better (aesthetics) graphics, which I believe would further enhance the impact of your paper.

Author Response

Reviewer 1

Comments

I would like to see the results organized differently, and/or with better (aesthetics) graphics, which I believe would further enhance the impact of your paper.

RE: Thank you. We have improved the graphics for consistence and better display.

Reviewer 2 Report

Comments and Suggestions for Authors

In this manuscript, the authors studied the immune response of farmed immunized Atlantic salmon to bath and intraperitoneal Moritella viscosa challenges and evaluated the immunogenicity of M. viscosa cell components, the antigenicity of the different M. viscosa components (bacterin and exudates'(complete bacterin), bacterin cell, IROMP, OMVs) in Atlantic salmon was evaluated in terms of IgM titers, peripherical and spleen IgM+ cell populations. The results are interesting; however, some issues as below should be resolved before publication in Vaccines.

1. More details about IROMP and OMVs might be described in the Introduction. Moreover, “outer membrane vesicles (OMVs)” should be defined at the first appearance in the text (line 71), but not in line 74 and 230, similarly, IROMPs and IRON should be defined at their first appearance in the text.

2. The background about the immunized Atlantic salmon vaccinated with ALPHA JECT micro 80 IV should be described, for example, how long after the fish is immunized is used for challenge assay?

3. In line 241-242, “six 500 L tanks, each containing 45 fish”, while in Line 244-245: “In each tank, 15 fish were intraperitoneally boosted with their respective vaccine, and 15 were injected with PBS mixed with carbigen-adjuvant and used as the control group (Figure S3).” why just 15 fish were boosted with the vaccine or PBS?otherwise, 15 fish is not enough for blood sampling from 6 fish every two weeks (2, 4, 6, and 8)?

4. To accurately evaluated the effectiveness of the ALPHA JECT micro IV, it is suggested to add an experiment of bacterial infection in unvaccinated Atlantic salmon and test its mortality rate as a control for bacterial infection in vaccinated Atlantic salmon.

5. There is an issue with the sequence of 2.3 and 3.2. If it is to verify the flow cytometry results, it should be placed after the flow cytometry experiment.

6. The description in Lines 291 to 294 are not about the result of this article, the reference should be removed to Introduction or Discussion.

7. In Discussion, there were 21 paragraphs, it is too much. it is recommended to combine some paragraphs into one according to the discussed topic, for example, several paragraphs are about IgM.

Author Response

Reviewer 2

  1. More details about IROMP and OMVs might be described in the Introduction. Moreover, “outer membrane vehicles (OMVs)” should be defined at the first appearance in the text (line 71), but not in line 74 and 230, similarly, IROMPs and IRON should be defined at their first appearance in the text.

RE: Thanks for your comments. We have expanded IROMP and OMVs background in the introduction and defined them at first appearance.

Lines 63-69

The pathogen's capability to grow in an iron-restricted environment led to the synthesis of Iron-Regulated Outer-Membrane Proteins (IROMP), which have been proposed as important antigens that protect against bacteria [15]. Their interaction with the host immune system makes them suitable candidates for vaccine development [16]. Additionally, bacterial outer membrane vesicles (OMVs) contain protective antigens in their structures and can be used for vaccines as well [17-21].

  1. The background about the immunized Atlantic salmon vaccinated with ALPHA JECT micro 80 IV should be described, for example, how long after the fish is immunized is used for challenge assay?

RE: That information has been added to the manuscript per your suggestion.

Lines 194-196:

Atlantic salmon were vaccinated with ALPHA JECT micro IV when they reached a weight of approximately 60-80 g on October 16, 2021, six months before challenge.”

  1. In line 241-242, “six 500 L tanks, each containing 45 fish”, while in Line 244-245: “In each tank, 15 fish were intraperitoneally boosted with their respective vaccine, and 15 were injected with PBS mixed with carbigen-adjuvant and used as the control group (Figure S3).” why just 15 fish were boosted with the vaccine or PBS?otherwise, 15 fish is not enough for blood sampling from 6 fish every two weeks (2, 4, 6, and 8)?

RE: Thank you very much for your comments. We apologize for the written mistake and the sentence has been corrected. Each tank contained 45 fish, and we had 6 tanks, so we certainly had enough fish to experiment.

  1. To accurately evaluated the effectiveness of the ALPHA JECT micro IV, it is suggested to add an experiment of bacterial infection in unvaccinated Atlantic salmon and test its mortality rate as a control for bacterial infection in vaccinated Atlantic salmon.

Thank you very much for your suggestion. While a challenged control (unvaccinated group) was not included in our study, the high mortality rates in unvaccinated salmon facing M. viscosa challenge have been well-established in prior research (Karlsen et al., 2012, Lovol et al., 2009, Karsen et al., 2017, …). The pathogenicity and massive mortality rates, reaching up to 95.5% even with a lower dose of M. viscosa (105 cfu/ml), in unvaccinated salmon were reported in previous studies. This bacterium is not only virulent in Atlantic salmon but also in lumpfish and the results reported by our labs in the previous/ and on-going studies declare the pathogenicity of this strain. The absence of a challenged control group is justified based on the extensive evidence from previous studies confirming the expected outcomes in unvaccinated salmon facing this particular challenge. More importantly, our focus was specifically on haemato-immune response of non-boosted and boosted Atlantic salmon following the challenge and post boost immunization. Additionally, due to animal ethics and protocol guidelines from the Institutional Animal Care Committee and the Biosafety Committee at Memorial University, and the risk of high mortality, the repetition of this experiment is not within our scope of approved procedures.

  1. There is an issue with the sequence of 2.3 and 3.2. If it is to verify the flow cytometry results, it should be placed after the flow cytometry experiment.

RE: We reorganized the text and figures based on your suggestion.

  1. The description in Lines 291 to 294 are not about the result of this article, the reference should be removed to Introduction or Discussion.

RE: We have considered your suggestion and removed the description.

  1. In the Discussion, there were 21 paragraphs, it is too much. it is recommended to combine some paragraphs into one according to the discussed topic, for example, several paragraphs are about IgM.

RE: We merged those paragraphs based on your advice.

Reviewer 3 Report

Comments and Suggestions for Authors

The authors of this manuscript studied the immune response of farmed immunized Atlantic salmon to bath and intraperitoneal Moritella viscosa challenges and evaluated the immunogenicity of M. viscosa cell components. This seems a thoroughly performed study and well presented except for one thing. When doing a challenge study the authors ought to have a challenged control which is missing.

L 99: centrifuged at 6000 rpm. Please use relative centrifugal force (xg) which is specific.

Check the whole manuscript for this minor error.

L 217+218: 10% carbigen carbomer-based adjuvant. I do not know this adjuvant. Can you more specifically indicate the nature of this adjuvant, is it oilbased or not?

L 237:All latin species named should stay in italics.

L 244: slip of the pen mistake

Author Response

Reviewer 3

Comments and Suggestions for Authors

The authors of this manuscript studied the immune response of farmed immunized Atlantic salmon to bath and intraperitoneal Moritella viscosa challenges and evaluated the immunogenicity of M. viscosa cell components. This seems a thoroughly performed study and well presented except for one thing. When doing a challenge study, the authors ought to have a challenged control which is missing.

RE: Thank you very much for your suggestion. If you are referring to figure S5. We modified this figure per your advice because we had a control group (non-boosted) that received PBS+ADJ and the mortality rates was added for this group. Please consider the revised supplementary figures. If you are referring to the unvaccinated animal as a control group. Provincial and federal regulations make extremely difficult to obtain unvaccinated industrial fish.  While a challenged control (unvaccinated group) was not included in our study, M. viscosa J311 strain used in this study is virulent in Atlantic salmon as was reported previously by our group and others, but also in lumpfish.

.

Figure S5. Survival proportions were compared between boosted and non-boosted (PBS+ADJ) Atlantic salmon after ip and bath challenges with M. viscosa. A group of animals was not subjected to any challenges.

L 99: centrifuged at 6000 rpm. Please use relative centrifugal force (xg) which is specific. Check the whole manuscript for this minor error.

RE: Thank you. We converted it as you suggested. It was replaced by 4200 x g

L 217+218: 10% carbigen carbomer-based adjuvant. I do not know this adjuvant. Can you more specifically indicate the nature of this adjuvant, is it oilbased or not?

RE: It is a terminally sterilized, carbomer-based (Carbopol 934P) adjuvant suspension containing a proprietary emulsified component and is free of animal origin ingredients. Its milky-white appearance creates a smooth, uniform mixture when added to the vaccine.

L 237: All latin species named should stay in italics.

RE: Thank you. We considered your comments.

L 244: slip of the pen mistake

RE: We apologize for the written mistake and the sentence has been corrected.

Round 2

Reviewer 2 Report

Comments and Suggestions for Authors

The issues have been resloved.

Author Response

Thansk

Reviewer 3 Report

Comments and Suggestions for Authors

What I mean is an unvaccinated group, but since it is extremely difficult to obtain unvaccinated industrial fish, we have to live with what we get.

Author Response

Thanks